# Hidden Person Makes Dialogue Present: The Place of *It* in the System of Dialogue According to Cohen, Buber and Rosenzweig

**Ilya Dvorkin**

International Center for University Teaching of Jewish Civilization, Hebrew University of Jerusalem, Jerusalem 91905, Israel; idvorkin@mail.ru

**Abstract:** The philosophy of dialogue is often presented as an attitude towards the world within the framework of the relationship "*I* and *Thou*". Martin Buber represented this approach in his works. Meanwhile, for other philosophers, especially Cohen and Rosenzweig, dialogue is unthinkable outside of a more complete system of person relations, which also includes He, She, They, We. A particularly sharp controversy unfolded between Rosenzweig and Buber around the place of *It* in the dialogical process. Rosenzweig not only criticized Buber's belittling of *It* and ignoring the deep connection of the *I–Thou* and *I–It* relations, but also built his own philosophy of the third person, which is an important element of his philosophical system as a whole. In particular, Rosenzweig showed the extraordinary role of the *It* in the construction of language. Rosenzweig's concept of *It* not only challenges Buber's *It*, but also echoes Freud's *Id*. Rosenzweig's philosophy can be seen in many respects as an attempt to harmonize the relationship between *I* and *It*, i.e., between the selfhood of a separated person and the closeness of the world completed in itself.

**Keywords:** dialogue; philosophy of dialogue; person; I; Thou; He; She; It; Id; We; speaker; present; hidden; relationship; correlation; logic of origin

*To Paul Mendes-Flohr*

*Intellectual life in Germany at the beginning of the 20th century reached its peak. This applies to most areas of culture, including philosophy. The participation of Jews in common cultural processes was also maximal. It was hard to imagine that several years would pass and a catastrophe would break out. Now, after the catastrophe, it is very important to study this era and retrieve the results achieved then for the sake of moving towards our future, which always be a future "after." When we turn to the time of the first third of the 20th century in Germany, the research of prof. Mendes-Flohr becomes an important reference point for us. This also applies to the main topic of our search—the origins of the philosophy of dialogue in the intellectual polemics of the early 20th century.*

## 1. "Cripple for an Opponent", Rosenzweig Criticizes Buber's Concept of *It*

The philosophy of dialogue in the eyes of many is associated with the opposition of the relationship *I–Thou* and *I–It*, which was introduced by Martin Buber. According to his concept, the dialogical principle consists in the personal *I–Thou* relationship, the experience of closeness and reciprocity, which opposes the subject-object relationship *I–It*. Meanwhile, Buber was not the only creator of the philosophy of dialogue, and it is not identical with his ideas. The opposition to *I–Thou* and *I–It* was taken up and criticized by many researchers and thinkers in the 20th century[1] and Buber's associate Franz Rosenzweig even before the publication of "I and Thou"[2]. After reading the manuscript, in September 1922, Rosenzweig wrote to Buber:

> "In your setting up the I-IT, you give the *I-Thou* a cripple for an opponent. < . . . > You make of creation a chaos, just good enough to provide construction material (Baumaterial) for the new building". (Rosenzweig 1979, pp. 824–27)[3]

Rosenzweig shares Buber's understanding of the importance of the I–Thou relationship. At the end of the letter, he declares: "Deep in my soul, behind the drawn curtains, I am still only interested in 'I' and 'Thou'" (Rosenzweig 1979, p. 826). The interpersonal relationship expressed by the pronoun of the second person remains the common basis of thought among all philosophers of dialogue. Rosenzweig's criticism is aimed at underestimating of the It and separating it from the I–Thou relationship. As Rosenzweig notes, already the use of the neuter for this third–person pronoun, is intended to devalue and discredit it. It is not It, but He should correspond to I and Thou—proclaims Rosenzweig (Rosenzweig 1979, p. 825).

According to Rosenzweig, Buber was so impressed with the discovery of the second person that he denied the third person. As Rosenzweig explains, this was insanity, which posed a huge danger to humanity. Rosenzweig notes that something similar happened to Hermann Cohen. In the preface to his writings and in private correspondence, he writes that Cohen was the first among the dialogists of the early 20th century to discover *Thou*, but this was not appreciated by the philosophers of that time. (Rosenzweig 1984, p. 181). Although Rosenzweig recognized Cohen's priority in discovering *Thou* as the principle of the new philosophy, he believed that Cohen's system did not fully express this philosophy. In this letter, Rosenzweig noted that having discovered *Thou*, Cohen tried to rebuild his entire, already completed philosophical system in order to include personal relationship into it (Rosenzweig 1979, p. 825)[4]. Buber in *The History of the Dialogical Principle* also recognized Cohen's priority, emphasizing that

> "[i]t is significant that the vision of the Thou is renewed first of all by the Neo-Kantian thinker Hermann Cohen". (Buber 2002, p. 252)

Buber was surprised that such a systematically thinking philosopher as Cohen becomes the discoverer of *Thou*. However, this very perplexity is the evidence that Buber did not see the connection between the dialogical principle and the entire systematic philosophy, which Cohen represented.

Unlike Buber, Rosenzweig understood perfectly well that the discovery of the dialogical principle should not push away the old classical philosophy, but rather, overcome and integrate it. Therefore, Rosenzweig speaks of the weakness of the position of Buber (and Ebner) who, emphasizing the importance of *Thou*, did not explore the entire system of the three persons. This also led them to ignore the importance of the first person plural *We* (Rosenzweig 1979, p. 826). According to Rosenzweig, only the awareness of the inseparable connection *Thou* and *It* allows us to build a new system of philosophy that will overcome the alienation of thinking defined by subject-object relationships.

## 2. Collision of *It* in 1921–23: Buber, Freud and Rosenzweig

To appreciate the significance of Rosenzweig's controversy with Buber about *It*, we must consider in detail the study of *It* in European philosophy and psychology.

Researchers have long noticed the fact that the release of two famous works of the 20th century with almost the same title, *I and Thou* by Buber and *The Ego and the Id* by Freud, both took place in 1923 (Kaufman 1980, p. 241). Since in English and in some other languages the personal pronoun of the third person does not have a neuter gender, the contrast between Buber's *Es* and Freud's *das Es* is not perceived as sharply by their readers in English, since they are translated differently[5]. Buber has *Es*, and Freud has *Id*. Meanwhile, It (as *das Es*) has a rich tradition of use in German philosophy. Freud directly points to the origin of Id (Es) from Nietzsche[6]. Not only Freud, but also Cohen, Rosenzweig and Buber relied on the same German philosophical tradition of treating *Es/It*. In this regard, the use of the same range of concepts by these authors is not accidental[7].

In this context, the difference between the interpretation of *It* in German philosophy and in psychology at the beginning of the 20th century is important. If we compare the use of *It* in Buber and his closest colleagues Cohen and Rosenzweig, we see that both of them prefer to use the personal third person pronoun of the masculine gender *He*, while

often using *It* as an ersatz version of *He*. Thus, in his *Ethics of Pure Will*, Cohen makes clear distinctions between the philosophical meanings of the second and third person pronouns (Cohen 1904, p. 234). Rosenzweig, as we noted, examines in detail the relationship of all three persons, and even raises the question of the gender of the third person. For him, the third person can be *He, It* or *She*. In his commentary on the history of the Fall, he directly correlates *It* with the serpent, which in turn, is associated with the animal instinct. The collision of relations between *He, She* and *It* is very clearly visible in this text. In describing human behavior after the Fall, Rosenzweig writes:

> "The man hides, he does not answer, he remains mute, he remains the Self as we know it <...> it is a He-She-It that comes out of the answering mouth; the man objectifies himself in order to become "the male human"; the woman, for her part, totally objectified as woman who is 'given' to the man, is the one who did it, and she then throws the guilt on the last It: it was the serpent". (Rosenzweig 2005, p. 189)

Thus, Rosenzweig's *It* unexpectedly approximates the Freud's *It/Es*, although Rosenzweig's text was written several years before the publication of Freud's famous treatise[8]. For Rosenzweig, *It* as an animal inhuman attitude combines the cosmic and psychological aspects.

In the first part of *The Star of Redemption*, in the description of the silent primordial world, the properties of the third person are spelled out in great detail. The third person is both a mythological god and a plastic cosmos, and even the self of a tragic hero. It turns out that, according to Rosenzweig's concept, the problem of correlation between *It* as the external objective world (Buber) and *Id* as the inner hidden world of the Self (Freud) is already anticipated.

In the above-mentioned letter, Rosenzweig warns Buber of the enormous danger of reducing the third person to *It* and of consequently separating *It* from the second person. Nevertheless, Buber insisted on the absolute priority of *Thou* and on discrediting *It*. When translating his book into Hebrew, which, like English, does not have a neuter pronoun in the third person, Buber agreed to use the Hebrew word "הלז" (*laz*) to translate the word *It*. This word from the biblical text is an expression of contempt and alienation[9]. Comparison of the concept of *It* in Buber, Freud and Rosenzweig brings to the fore the question of the relationship between the external physical reality of the cosmos and the internal psychological reality of life and the human soul. In the 19th century, understanding of this relationship radically diverged, despite the Kantian attempt to link these two realities to one another. The splitting of the meanings of *It* in Buber and Freud entailed a complete split of the inner and outer conceptual worlds. Paradoxically, the principle of subjectivity in science in the 19th and early 20th centuries gave rise to the incommensurability of "the starry sky above me and the moral law within me" (Kant 2002, p. 203). Buber's discrediting of *It* means a recognition of the complete failure of attempts to build a worldview that would harmoniously combine the external and internal. Meanwhile, by separating *Thou* and *It* into different poles of his philosophy, Buber more than anyone else problematized the *It*. The sharpness of Rosenzweig's criticism of Buber's concept of *It* calls for a philosophy in which all three persons will act in unity. In such a philosophy, the external objective and internal psychological plans of the third person would be studied with the same thoroughness. We can discover the first step in this direction in the philosophy of persons in Cohen and Rosenzweig, and their analysis will become an important part of our research. However, before embarking on this task, let us consider how the concept of three persons was formed in the mystical and philosophical literature.

### 3. The Speaker, the Hidden and the Present. Three Persons of the Mystical Tradition

Buber's insufficient assessment of *It* was all the more surprising since he had serious reasons to investigate the relationships of all three persons more thoroughly and not reduce everything to *Thou*. The fact remains that in Hasidic literature, the study of which Buber devoted most of his life, considerable attention was paid to the problem of the relationship

of the three persons. The most famous author who examined this relationship repeatedly and in detail was Rabbi Zadok ha-Kohen Rabinowitz of Lublin (1825–1900), who was undoubtedly well known to Buber. Zadok ha-Kohen constantly emphasized the importance and unity of all three persons. Thus, in his book *Mahshavot Haruts* (Zadok ha-Kohen 1912, chapter 5), he writes: "And the three crowns are the crown of the Torah, the Priesthood and the Kingdom against three levels: *I*, *Thou*, *He*."

Categories of personality are expressed by the three pronouns *I*, *Thou*, *He*. They are a constant motive in the works of Zadok ha-Kohen. In this case, speaking of their unity, he compares them with the well-known principle of the unity of the three crowns (Torah, Priesthood and Kingdom) formulated in the Mishnah (Pirkey Avot, 4, 13). At the same time, Zadok ha-Kohen emphasized the importance of "the level of *He*": "But the crown of the Torah, which is above them all, is the level of *He*, which expresses the depth of the concealment contained in those who are attached to *Him*" (Zadok ha-Kohen 1912, chapter 5). Of course, Zadok ha-Kohen is by no means the first to address the mystical concept of persons. In this regard, he constantly refers to his main source—the *Zohar*. The three mystical persons are important elements not only in Jewish mysticism, represented here by Zadok ha-Kohen, but also in Islamic mysticism, in the doctrine of Sufism. Both Jewish and Muslim mysticism use the designation of persons in medieval Hebrew and Arabic grammar. The first person *I* is characterized as "speaker" (or the one who speaks) (מדבר), the second person *Thou* as the one who is "present" (נוכח) and the third person *He* as the one who is "hidden" (נסתר). For this reason, Zadok ha- Kohen designates *Thou* as recognition of the one who is present, and *He* as the expression of a "secret hidden knowledge."

The conclusion that we can draw from this short review of Jewish and Muslim mysticism[10] is that all three persons have significance, they are parts of one system. It was not a philosophy that started to study this system, but rather, religious mysticism.

### 4. From Subjectivity to Dialogue. Three Persons in the History of Philosophy

The question of including personal pronouns as basic concepts in philosophy has been discussed for over a hundred years. H. Cohen notes that the first person plural pronoun plays an essential role in Plato, but not as a concept, as a position statement (Cohen 1904, p. 199). According to Cohen, personal pronouns could not become essential concepts in ancient Greek philosophy due to the lack of conceptualization of the idea of a Man. According to Cohen, the first one who introduces personal pronouns as a matter of philosophical discussion is Philo of Alexandria. Philo tried to combine Greek philosophy and biblical thought (Cohen 1904, p. 200).

Of course, we owe Descartes the direct introduction of *I* into philosophy; we also ascribe him an understanding of the *I* as the privileged place of subjectivity. Yet, his discovery of the philosophical importance of the *I* was prepared by previous developments. By the 17th century in Europe, the formation of the position of an individual thinker, and thus, of the "I think" was taking place; such an individual thinker sees the world through the eyes of his/her own mind.

In the picture of the world formulated by Descartes, the basis of being is located outside of a human, whereas the basis of knowledge is the self-contained *I* of a human.

Although Cartesian dualism was trying to overcome Spinoza, the main discovery of Descartes' philosophy—*I* as the center and fulcrum of the process of cognition—becomes the basis of the European picture of the world.

Meanwhile, the transformation by Descartes of the personal pronoun *I* into a concept meets resistance in Kant. Introducing the concept of transcendental subject, Kant tries to get away from *I*, which seems to him too empirical (KrV, A 429/B 545). At the same time, the personality of *I* finds an unexpected ally in German philosopher Jacobi, who was an important critic of both Spinoza and Kant. To him, apparently, belongs the honor of the philosophical discovery of the concept *I–Thou*, which, in contrast to the mediated subject-object relationship, expresses an immediate sense of relationship.

It is not surprising that Buber starts his study of the philosophy of dialogue in his work "The History of the Dialogical Principle" (Buber 2002, p. 250) with Jacobi, whose position is extremely close to him. Even when moving away from this position, Buber consistently considers the process of turning *Thou* into the foundation of a new philosophy. Under the influence of Jacobi, and to a large extent, in opposition to him, Fichte's philosophy was formed in which *I* is the central point.

In his work, Buber tries to restore the entire sequence of transformation of the personal pronoun *Thou* into a fundamental concept. However, pronouns of the third person—*He*, *She*, *It, They*—do not interest him. Buber strongly identifies the dialogical principle with the *I–Thou* relation. In this regard, the transition from Jacobi to two such different thinkers as Feuerbach and Kierkegaard is also understandable. The only thing that unites them is that they are in sharp opposition to Hegel's absolute idealism. But they criticize Hegel from different positions. Kierkegaard reveals the limitations of Hegel's fusion of philosophy with faith and the absorption of revelation by reason (Mendes-Flohr 1989, pp. 359–61). Feuerbach, on the contrary, desacralizes the religious concept of man, giving him a materialistic interpretation. At the same time, the total impersonality of Hegel's philosophy evokes a sharp protest from both of these philosophers. Rosenzweig expands the range of sources and adds Schopenhauer and Nietzsche to them (Rosenzweig 2005, pp. 11–15). Both reveal the irreducibility of man to thinking and reveal the importance stress and the uniqueness of the human personality. All these authors proposed revolutionary ideas and the destruction of stereotypes, but this is especially characteristic of Nietzsche. He actually completely rethinks the main concept of European philosophy—human. From a given, human turns into a goal, from static reality into process. The concepts developed by Nietzsche—*I, Self, It, Super-Man*—become the basis of philosophy and psychology the 20th century.

All these philosophers, from Jacobi to Nietzsche, did not engage in systematic philosophy, but rather, tried to construct philosophical doctrines aimed at life itself. In this regard, the steps they made in comparison with Kant, Fichte, Schelling and Hegel are based in their worldviews and did not become the basis of philosophical systems. Therefore, the real philosophical development of the concept of person remained unfulfilled and was waiting in the wings. As stated by both Buber (2002, pp. 252–53) and Rosenzweig (2005, pp. 27–29), its time had come in the philosophical system of Hermann Cohen.

## 5. Three Persons and Their Place in the Philosophy of H. Cohen

Unlike the mentioned philosophers of the late 18th and second half of the 19th century, Hermann Cohen was a systematic philosopher who relied on the philosophical texts of his predecessors and built his own philosophical system on their foundations. This quality became the most important reason for his outstanding role in the formation of new thinking; and at the same time, led to difficulties in understanding of his philosophy in the middle of the 20th century. Meanwhile, it was Cohen who summed up the development of critical German idealism (Poma 1997) and prepared the ground for the birth of a philosophy of dialogue (Dvorkin 2020).

Although we find a systematic study of person relations in the second and fourth parts of Cohen's philosophy in his *Ethics of Pure Will* and *Religion of Reason*, we must start the research with the first part of his system—the logic. This is exactly what Rosenzweig does in his own development of the philosophy of dialogue. Rosenzweig's starting point is Cohen's logic of origin (*Ursprung*) (Rosenzweig 2005, pp. 27–29). Rosenzweig draws on one of the most important ideas of the *Logic of Pure Knowledge* (Cohen 1902): the realization that the subject of knowledge is neither an abstract thing-in-itself nor a phenomenon that does not reflect things in themselves, but the process of generation, i.e., the origin of the phenomenon (*Ursprung*). Thinking, according to Cohen, is not a reflection of an external or internal reality given to us, but a correlative dynamic process of generation in which this reality is formed. It follows that both space and time, and all categories of mind are not a static picture, but a dynamic process.

Based on these ideas, in his *Ethics of Pure Will*, Cohen studied the structure of the self-consciousness of *I* and came to the conclusion that the dynamic origin of *I* is *Thou*—the partner with whom *I* establishes a relationship. Cohen emphasizes:

> "that *I* cannot conceive *I* without conceiving *Thee*. So, in self-consciousness, the Other has transformed into Thou in a duality with *I*. So far as self-consciousness means the unity of will, it has to create the union of *I* and *Thou*". (Cohen 1904, p. 235)

For Cohen, the *I–Thou* relationship is not an abstraction that describes reality, but the process realized in a state of mutual responsibility. This process is expressed in his concept of legal contract:

> "Always implying at least two parties in the conversation, the contract then transfigures the claim (Anspruch) into an address (Ansprache). And therefore the Other to I transforms into Thou. Thou is not He. He would be the Other. He is likewise in danger of being treated as It. Thou and I essentially belong together". (Cohen 1904, p. 235)

On the one hand, the contract is only an example of the relationship between *I* and *Thou*; on the other hand, it expresses the very idea of mutual obligations, which is characteristic of dialogue. From the very beginning, Cohen does not reduce the *I–Thou* relation to an existential experience. As a system of correlative relations between *I* and *Thou*, a contract must always have an object. There is always something expressed in the third person—*He*, *It*, *She*—around which the relationship between *I* and *Thou* is built. But it is not these circumstances that become the origin of *I*. The origin is *Thou*. Cohen discovers the difference between the second and third persons in the comparison of objectivity and personality:

> "Yet, up until now, the kind of human peculiarity inherent in the *Thou* has not been positively determined; apparently, it is the personality that is brought to light more through the *Thou* than through the *He*. The *He* is more subject to neutrality, which makes it hardly distinguishable from the *It*". (Cohen 1972, p. 16)

This does not mean, however, that the world of the third person is an external objective reality and the world of the second person is the world of personal relationships, as Buber suggests. For Cohen, the ethical dimension, which is also expressed in the second person, nevertheless requires a third person. Ethics requires some objectivity, the relationship between *I* and *He*, who then passes that objectivity into *Thou*. Moreover, the objective world, expressed as it is in the third person, is drawn into the world of interpersonal relations. As if responding to the Kantian demarcation of the world of objective knowledge ("the starry sky above me") and the world of subjectivity ("the moral law within me"), Cohen notes that this separation and the resulting totality of the object is a consequence of ignoring the second and third person singular and the first person plural:

> The universality (*Allgemeinheit*) cannot be confused here with the totality (*Allheit*). For, the totality is represented by *I* alone. In the case of the universality, on the other hand, it all depends on the development towards *Thou* and towards *He*, and *He* in turn is transformed into *Thou*; and all *He* are revealed when they are drawn together into *We*. (Cohen 1904, S. 260)

Thus, for Cohen, the three persons collectively form a system of dynamic relations in which the reality of the world unfolds in the universality of human dialogue. The intrinsically interpersonal character of personality in self-awareness makes it necessary to introduce another important pronominal concept, namely, *We*. As a result, the whole reality becomes the realization of the process of interpersonal relations.

In the already mentioned letter, Rosenzweig wrote that Cohen discovered something his own system could not cope with, so he had to revisit the system and create the philosophy of religion for this system's part. It seems that this assessment is only partly correct. Indeed, Cohen's ideas evolved throughout his life. But this does not negate their continuity and internal connection. His philosophy of religion cannot be imagined without his

ethics and aesthetics. But without *The Religion of Reason*, Cohen's system also would not be complete. No wonder the concept of interpersonal relations in *The Religion of Reason* differs significantly from that in *The Ethics of Pure Will*. However, this difference lies in a significant development of ideas, and not in the formation of a new system. Indeed, in *Religion of Reason*, Cohen relies on his theory of interpersonal relations. At the beginning of the book, he reiterates the conclusions of his ethics:

> < . . . > yet another mediation is needed besides the one required between the I and humanity. Besides the I, and distinct from the It, there arises the He. Is the He only another example of the I, which is therefore already established by the I? Language alone protects us from this mistake; language sets up the Thou before the He. Is the Thou also only another example of the I, or is a separate discovery of the Thou necessary, even if I have already become aware of my own I? Perhaps the opposite is the case, that only the Thou, the discovery of the Thou, is able to bring myself to the discovery of I, to the discovery of the ethical knowledge of my I". (Cohen 1972, pp. 14–15)

But then he builds a new theory of *I–Thou* not based on ethics. The outstanding feature of Cohen's new concept is his rethinking of the very nature of the relationships between a human and another fellow human. If in ethics these relationships are articulated in the language of responsibility and mutual obligations, in *Religion of Reason* human relations acquire an independent meaning, now rooted in religion. Here instead of an ethical concept of a human as my partner (*Nebenmensch*, literally "a man *next* to me"), the concept of a human as my fellow (*Mitmensch*, literally "a man *with* me") appears. The main theoretical construct that forms the concept of *Mitmensch* is the construct of "getting closer" (*Annäherung*, literally "getting into proximity") (Cohen 1972, pp. 20–21). The other human is not only a partner; he is my proximity, my neighbor (Nächste). For Cohen, the relationship of closeness, proximity and intimacy becomes the basis for a new understanding of interpersonal relationships. Closeness as a characteristic of another person expresses the presence of the divine in him. Participation in the life of my fellow man, a man who is simply given to me in my proximity, turns out to be one of the most fundamental human traits. Cohen articulates this trait in his notion of *religion*. However, it does not mean that interpersonal relations for Cohen are reduced to *I* and *Thou*. His concept of *closeness*, like his ethical and legal concept of *responsibility*, requires an appeal to a third person. For Cohen, the other person is not a pretext for existential experiences, but the reason for moral action.

### 6. The Silent Primordial World of Rosenzweig. Cosmos and Self as Prerequisites for Dialogue

Based on the ideas of Hermann Cohen, Rosenzweig builds his own philosophy of dialogue[11]. Before considering in detail his concept of persons and interpersonal relationships, we must examine his understanding of the impersonal world, which he explains in the first part of his *Star of Redemption*. This world is described by Rosenzweig as "silent" (Rosenzweig 2005, pp. 73, 85), but it is viewed from the perspective of a new speaking think. Thus, the transformation that occurs in the second and third parts of the *Star of Redemption* is already intrinsic to the first part. Although it deals with closed elements in which personality is dissolved, we know that these elements can open up. Rosenzweig speaks of two ways of such an opening-up or disclosure—from the inside and from the outside (Rosenzweig 2005, p. 28). The first works already in the first part of *The Star*, and the second in the second.

In the first part, following H. Ehrenberg's footsteps, Rosenzweig formulates three meta-sciences for each element—metalogic for the world, metaethic for man and metaphysics for God. Their task, as Rosenzweig explains, is to explore Something, not Nothing, of the world, man and God. This reality of Something is elusively small in relation to Being in its opposition to Nothing. Why is the silent primordial world as important to Rosenzweig as the speaking world of the second part? This question is easy to answer: since Rosenzweig finds the foundations of speech in the elements of the silent primordial world.

The static plastic cosmos becomes the dynamic world of creation; the self-contained self of the tragic hero is revealed as the speaking personality of revelation; and finally, the mythological gods contained within themselves are revealed in the personal God of creation, revelation and redemption. The logical primeval words "Yes", "No", "And" explored in the first part become the root-words and sentences of living language in the second part.

Thus, the silent primordial world of the first part of Rosenzweig's *The Star of Redemption* surprisingly combines the *It* (*Es*) of Buber and *Id* (*das Es*) of Freud. The external static world of Buber's *It* corresponds to Rosenzweig's plastic cosmos in the first book of the first part of *The Star of Redemption*. Rosenzweig deploys both the *It* (*Es*) of Buber with the *Id* (*das Es*) of Freud. However, in the second book, Rosenzweig goes beyond the objectivist view of the world. Hidden from himself, Freud's *Id* now corresponds to the "self" of the tragic hero. Rosenzweig begins his exploration of the soul of the tragic hero with what he calls *negative psychology* (Rosenzweig 2005, p. 71), following Rosenstock-Huessy, who, in his *Practical Knowledge of the Soul* (Rosenstock-Huessy [1924] 1988), regards the self of the "closed soul" as a disease that requires healing. Rosenzweig describes human speech as the birth of soul (Brasser 2004). However, this birth of soul, which is described in the second part, has its origin in self, which is described in the first part. Similarly, the plastic cosmos of the first part turns into the created world in the second part. This circumstance helps to understand Rosenzweig's remark in his letter to Buber: "*It* is not dead. *It* is created "(Es ist geschaffen) (Rosenzweig 1979, p. 825). If Freud's *Id* can be correlated with something in Rosenzweig, this would be just the *Self* of the first part of *Star of Redemption*. Their *It* is understood as "dead". However, in the second part of the book, the self finds its expression in the idea of third person. So here, "It is not dead"[12].

In the second part, all the elements come out of isolation, acquire movement and begin to interact with each other. As Mendes-Flohr notes, holistic knowledge, according to Rosenzweig, is born from the interaction of knowledge and faith, which represent the objective and subjective aspects of reality. The meeting of a *new philosopher* with a *new theologian*, which is described at the beginning of the second part (Rosenzweig 2005, pp. 116–17), becomes a meeting of the subjective and objective aspects of the personality that generates new thinking (Mendes-Flohr 1989, p. 361). Meanwhile, the meeting of the objective and subjective dimensions of reality also signifies the meeting of the cosmic and psychological aspects of the third person. The transition from silence to speech is rooted in the elements of the silent primordial world. As Rosenzweig emphasizes, the words of thinking turn out to be the basis of the root-words and sentences of the language. But in the dynamic reality of creation, revelation and redemption, the elements interact with each other.

### 7. *It* in the System of Dialogue. Generation of Language in the Philosophy of Rosenzweig

In the second part of *The Star of Redemption*, Rosenzweig moves from the closed world of thinking to the open and dynamic reality of language. However, in the first book, speech is not yet a dialogue, and impersonal demonstrative pronouns still dominate language. This reflects the fact that the first book deals with the world of creation, and the language described in it is the language of creation (Askani 2004). Dialogue and interpersonal relationships only appear in the second book, which describes revelation. In the *Logic of Creation* section, Rosenzweig carefully analyzes the difference in approaching things and matters in the philosophy of idealism and in the "new thinking." Idealism in its highest form in Kant and Hegel already implies that a thing as a phenomenon differs from a thing-in-itself. Hence, already in Kant, and especially in Hegel, a clear distinction is made between being (*Sein* in the sense of *being* per se) and existence (*Dasein* in the sense of being-as-a-phenomenon). However, idealism sees existence as a product of *being* per se. In Hegel, for example, *Dasein* turns out to be an expression of the identity of *Being* and *Nothing*. Rosenzweig compares this to the Neoplatonic concept of emanation (Rosenzweig 2005, p. 147).

However, created thing, as Rosenzweig understands it, is radically different from thing as it is of the classical philosophy. To describe created thing, Rosenzweig uses the same term, *Dasein*, as Hegel did; yet unlike Hegel, he finds the origin of a thing not in "nothing", but rather, in the fact that this thing has been created. What for Hegel is *nothing*, for Rosenzweig is the act of creation. Rosenzweig, as late Heidegger also does, plays with the meaning of the word Dasein, breaking it into two words Da-Sein, "here being", but the meaning of this game is different for Rosenzweig. By saying that the *thing* is here, he emphasizes its actual presence in relation to the person and even some autonomy. As Rosenzweig notes, the thing is already here, it precedes the human who perceives it (Rosenzweig 2005, p. 143). Rosenzweig's expression "*Schon-da-sein*" (being-already-here) is a manifestation of the creation of the world. However, this creation does not mean the dead static nature of reality. *Already here* is not a passive past that grasps thought, but the actual present or, more precisely, the past actualizing itself in the present.

In describing being-creation, Rosenzweig relies on Cohen's logic of origin, which considers being a correlative process. Things are not given; rather, they are assigned as the process of correlation with their origin. Thus, creation "in the beginning" means the continual creation of a thing at every moment. (Cohen 1972, p. 68). In this connection, Cohen recalls the words of the morning Jewish liturgy "וּבְטוּבוֹ מְחַדֵּשׁ בְּכָל יוֹם תָּמִיד מַעֲשֵׂה בְרֵאשִׁית" (in His goodness renews every day, continually, the act of creation). Rosenzweig uses the same words to describe creation (Rosenzweig 2005, p. 121). "Deed in the beginning" (Maase bereshit) from Talmudic times is a description of the creation of the world in the first verses of the Pentateuch. But this text does not use the word *creation*, but speaks of *renewal*. Cohen, and after him Rosenzweig, consider the renewal that occurs in each moment as the actualization of creation. This is the meaning of the expression *Schon-da-sein*. The already-here-being of the world means, for Rosenzweig, its creation. The world has already been created, but the perception of it as created requires the actuality of creation at every moment.

In this theory, Rosenzweig follows Cohen in many ways, but instead of correlation, he speaks of path. However, this difference is not as significant as it seems. The idea of path is based on the same logic of origin as correlation. It simply emphasizes even more vividly than Cohen's notion that reality has not a static but a dynamic character. For Rosenzweig, his meta-logical world, the plastic cosmos of the first part, turns out to be the origin for considering the created world of the second part:

> "We had accompanied the world in its self-configuration to the point where it seemed complete in itself, < . . . > Only the idea of Creation tears the world from its elementary enclosure and motionlessness and carries it away into the river of the All". (Rosenzweig 2005, p. 129)

At this point, Rosenzweig's new idea has emerged, and it is different from Cohen's concept. He states that the processuality of the world, found in the path, is expressed not in thinking, but in language and speech (Sprache). It is the activity of language that is realized in the present, and it is language that allows us to describe the world as created. The entire second part of Rosenzweig's book is based on the concept of the *grammatical organon* (Dvorkin 2021). In the first book, he explores the language of creation; in the second, the language of revelation; and in the third, the language of redemption. The language of creation describes the already present world, the cosmos, which has ceased to be a picture and has become an actuality.

We can now return to Rosenzweig's remark in his letter to Buber, "*It* is Created." As Rosenzweig emphasizes in his logic of creation, "the world is created." The reality of *already-here-being* is described by the variety of demonstrative pronouns; this is the reality of the third person. Dialogue appears in the second book in the language of revelation, but this in no way diminishes the language of creation. Its root word is Good, which Rosenzweig describes as a free predicative. In accordance with this thought, the property of predicativeness acts as the generative element for an entire system of grammar. Hence,

as Rosenzweig emphasizes, the third person, who has an advantage in the grammar of creation, becomes the most important for the formation of an entire system of language:

> "The passage through the pronoun establishes the thingliness of things which are, of course, due to this passage, straight off in the third person". (Rosenzweig 2005, p. 142)

The dialogic aspect of the language is revealed by Rosenzweig in the second book of the second part. Here soul's *self*, which is the subject of the first part of *The Star of Redemption*, turns into personality. The root word of the second book, the personal pronoun *I*, is not understood as a subject. Rosenzweig clearly distinguishes this *I* from the subjectivity of ontological philosophy. He repeats Cohen's assertion that a person becomes *I* only in relation to *Thou* (Rosenzweig 2005, p. 188) However, the first person, as seen by Rosenzweig, acquires language based on the third person. The object of the language of revelation is the created world of the first book. This makes understandable Rosenzweig's indignation at Buber's belittling of the world *It*. Without the created world of the first book, the revelation of the second book would turn into a pure mystification.

The process of redemption is expressed by the reconciliation of creation and revelation, i.e., third and first persons. According to Rosenzweig, this leads to the formation of the completed (or eternal) *We*. In a way, this interaction is similar to the process of reconciliation between *I* and *It*. Here, Rosenzweig agrees with his friend and opponent Rosenstock-Huessy. The healing of a soul occurs through language. The difference apparently lies in the fact that Rosenzweig's language not only heals the soul but opens it and provides it with the opportunity to make a dialogue about the world. This dialogue harmonizes the relationship between creation and revelation and thus resolves the conflict in the relationship between the first and the third person.

New thinking allows for a fresh look at the main problems of science. Thus, Rosenzweig writes in the first part about negative psychology, cosmology and theology. After the inclusion of speech in the present tense, our vision of the world radically changes. Rosenzweig's reasoning about the world, God and man in the second part can be seen as the outline of a new science. Although Rosenzweig remains primarily a philosopher in his book, he is not averse to the application of his ideas to concrete fields. But in general, this remains a task for the subsequent development of philosophy.

## 8. Philosophy of Dialogue as a Philosophy of Three Persons

It is clear from the above that, in accordance with the logic of Cohen and Rosenzweig, the separation of a third person from two other persons deprives interpersonal relationships of integrity and dynamics. Only a system with all three persons can act as the basis for any philosophical system. Each of them represents the special position, and their relationship has its own specifics.

Historically, the most developed is the first person position. In modern philosophy, it is defined as the position of *subjectivity*. Descartes does not use the term *subjectivity* in the modern meaning. Nevertheless, it is generally accepted that the philosophy of subjectivity from Descartes to phenomenology and existentialism of the 20th century is the basis of all modern philosophy. Moreover, subjectivity of the first person is also the basis of psychology and a number of other sciences. However, interpreting subjectivity without considering the addressing of human existence to *other* is flawed. The concept of inter-subjectivity solves problems. Rather, even if another person is a subject, and has subjectivity for himself/herself, but for me he/she is the interlocutor, the goal of aspirations, the origin of new knowledge, the source of love, are neither the subject nor the object. Hence the concept of the second person; in accordance with the medieval tradition, we call the second person "the *present*" (in the conversation) which, in the modern context of subjectivity, gives a fundamentally new dimension to philosophy. The discovery of the second person as "present" is the main achievement of the philosophy of dialogue of Cohen and Rosenzweig, Buber, Ebner, Bakhtin and Levinas. Nevertheless, the second person cannot be held as a fundamental philosophical concept without a third person, who, following the medieval

tradition, is thought of as "hidden in the conversation." Although the third person as an expression of being of thing in their always "hidden" participation in any conversation is well known to philosophy, its position in the framework of the dialogue of three persons has hardly been studied. Therefore, the study of the third, hidden person becomes one of the most important tasks of philosophy, psychology and theoretical linguistics.

Even the very fact that the concept of the second person is expressed by the noun of the third person (the one who is "present") testifies to their inseparability. This is evidenced by the reality of three persons as a linguistic universal. However, the need to talk about three persons follows not only from the properties of language, but also from logic. If we could not see the third person in the second person, we would not be able to distinguish it from the first. Addressing another person as *Thou* presupposes a certain distance and awareness of his/her difference from *the speaker*. To become *Thou*, it is necessary to contain some potential *He* or *She* therein.

We find an important development of the idea of the third person in Emmanuel Levinas. Like Rosenzweig, in opposition to Buber he insists on the asymmetry of interpersonal relationships (Peter et al. 2004). The third person for which Levinas uses the special term "l'illeité" (him-ness) reveals the asymmetry of this relationship (Levinas 1963, pp. 619–23). As he shows in *Autrement qu'être ou au-delà de l'essence* (Levinas 1974, pp. 201–2) I can put myself in the place of another, but a third person can never replace me. Thus, the third person expresses the asymmetry of proximity[13].

On the other hand, the speaker's attitude to the subject, i.e., the attitude of the first person to the third needs an interlocutor. Without *Thou*, I cannot identify an object as external and different from myself. Likewise, *I* cannot realize myself as the first person, unless there are other persons to whom my speech is directed.

The existence of three persons makes it possible to distinguish between thinking and speech. Thinking is always an internal process that unfolds within the framework of the relationship between the first and the third person. Speech is at least potentially addressed to the interlocutor and presupposes an *I–Thou* relationship. Rosenzweig pays great attention to the interpersonal nature of language and the implementation of linguistic phenomena in the present tense. This is also due to the fact that the second person, as present, exists always in the present tense. Meanwhile, thinking, speaking and acting phenomena are impossible without a third person.

The designation of the third person as hidden reveals its duality: an outward and inward orientation at the same time. The third person is addressed by both the first person and the second person. On the one hand, it forms the inner world of subjectivity, the direction of its addressing, its intentionality. On the other hand, the third person is the world of "my" compatibility with the second person, of "my" ability to address the mutual world. As we have seen, Cohen recognizes that the second person transforms the third person's world from totality to universality.

In this regard, it is extremely important to investigate the connection of the personal pronouns of the third person with the verb of existence and the auxiliary verb, the grammatical copula, "*is*". The meaning of the third person pronoun indicates an object that exists but cannot be addressed. We can say that the verb of existence "*is*" describes a reality even more alienated and even more independent from the speakers than the pronouns of the third person do. It is also suggestive that in Semitic languages the third person pronoun *He/She* functions as the copula "*is*".

Thus, the third person is directly related to the main theme of philosophy—the problem of being. This was perfectly understood by Buber, who declared the base-world of *I–It* to be the world of subject-object relations. The whole history of ontology can be understood as the history of the search for a third person. This concerns not only the concept of Being in Parmenides, Plato and Aristotle, but also the concept of Being as Dasein in Heidegger. Dasein is Being here in the third person. Heidegger was able to penetrate the mystery of the third person, but completely ignored *Thou* as the second person. Buber, by rejecting the third person, took the opposite route. However, even his *Thou*, as Rosenzweig pointed

out (Rosenzweig 1979, p. 824), was based on a great deception. And Freud's doctrine of the *Id* also contains deceit. By ignoring the connection between *It* and the *I–Thou* relationship (the personal relationship with another person), Freud turned the third person into a completely external and alien reality, the *Id*. Meanwhile, the very idea of psychoanalysis is based on the possibility of arriving at *It* through the *I–Thou* dialogue between the psychoanalyst and the patient. Freud overlooked the remarkable possibility of the theoretical justification of his methods within the framework of the concept of dialogue. This was partly done later by Lacan, but the gap between philosophy and psychology that arose in the 19th century did not allow him to bridge the gap between *It* and *Thou*.

Thus, we now see that the ideas of philosophy and psychology formed at the beginning of the 20th century could transform the dynamic relations of the three persons into a fundamental theoretical foundation. Apparently, Rosenzweig contributed to the development of this theme more than others. But he provided only an outline of research to be continued.

Continuing the direction Rosenzweig charted, we can develop the philosophy of dialogue as the study of the dynamic relations of all three persons, both singular and plural. This direction may further lead to the philosophy of the future. In such a philosophy, psychological and social phenomena will be described together as a single process. Language will be studied, not as a system of signs describing the external world, but as a dynamic system of interpersonal interactions. External and internal worlds will come into and be explored in a closer connection with each other, and the third person in this philosophy will play just as important a role as the first and second persons.

**Funding:** This research received no external funding.

**Conflicts of Interest:** The author declares no conflict of interest.

## Notes

1    This applies to both philosophical criticism itself (Kaufman 1978, pp. 31–38; Katz 1985) and psychological criticism (Frankl 1978, p. 66). For a review of the criticism of Buber's philosophy, see Zank and Braiterman (2022).

2    This letter was published in the collection of Rosenzweig letters (Rosenzweig 1979, pp. 824–27) and has been studied by many authors (Casper 1978; Horwitz 1978; Batnitzky 1999; Losch 2015).

3    English translation, Zank and Braiterman (2022).

4    Rosenzweig is not entirely consistent here. As D. Hollander notes, in developing his dialogic concept, Rosenzweig himself relies not so much on Cohen's latest religious works, but on his logic of origin, formulated in the first part of his philosophical system (Hollander 2008, p. 15).

5    About the untranslatable German *das Es* into English and French languages. See Cassin (2014, p. 294).

6    In his treatise "The Ego and the Id" (Freud 1960, p. 17), Freud refers to Georg Groddeck, who took It from Nietzsche. See Groddeck (1923).

7    See my article Dvorkin (2002) for a detailed analysis of this topic.

8    Already in October 1917, Rosenzweig noted the importance of das Es in his letter to Rosenstock-Huessy. See Letter 446, Rosenzweig (1979, S. 470).

9    The meaning of the word "*laz*" is This. The translator drew from fragment 2 Malachim 23, 17 "וַיֹּאמֶר מָה הַצִּיּוּן הַלָּז אֲשֶׁר אֲנִי רֹאֶה וַיֹּאמְרוּ אֵלָיו אַנְשֵׁי הָעִיר הַקֶּבֶר אִישׁ הָאֱלֹהִים" (And he said: what is this [הלז] monument that I see? And the inhabitants of the city answered him: This is the grave of the man of God.)

10    See Dvorkin (2018) for more details.

11    Dependence of Rosenzweig's philosophy on Cohen's ideas is well known and recognized by Rosenzweig himself. Researchers also write about it (Weiss Rosmarin 1974, pp. 61–62; Schwarzschild 1990, p. 335; Hollander 2008, pp. 13–39).

12    In his lectures, which Rosenzweig read at *Lehrhaus* in 1921–22 (Meir 2005, p. 44–45) even before his letter to Buber, he dwells in detail on the relationship between *I* and *It*. Rosenzweig explicitly states that *It* makes it possible to address *Thou* as *Thou*. (Rosenzweig 1984, p. 645) Buber also taught at this time at *Lehrhaus* and not only closely collaborated with Rosenzweig but discussed with him the problems of his not yet published book (Horwitz 1978). The criticism of Rosenzweig during this period radically influenced the formation of Buber's dialogical concept (Losch 2015). However, Buber did not accept a dynamic picture of the relationship of all three persons, which is characteristic of Rosenzweig.

[13]     Despite the similarity of these descriptions with Cohen's concept of proximity, Levinas studied completely new psychological, social and religious nuances here (Gibbs 1994, p. 241; Banon 2022, pp. 497–98).

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
