# Peer review of "Hidden Person Makes Dialogue Present: The Place of It in the System of Dialogue According to Cohen, Buber and Rosenzweig"

_religions, doi:10.3390/rel13060514_

Round 1

Reviewer 1 Report

a file is added

Author Response

I am very grateful to the reviewer for working with my article. I have substantially corrected the style and English grammar of the article.

I made correction according to all the comments of the reviewer, except for those that I did not understand: 328 take . off before (, 392 please clarify the sentence, 460 Mendes-Flohr

Reviewer 2 Report

Paper read very well. The theses presented in the text seem clear and have been presented in the paper. Properly chosen literature. Paper discusses an interesting problem at the interface between Bober's philosophy of dialogue and some threads of Rosenzweig's thought. All this is based on issues related to the tradition of Judaism. In my opinion, the paper is interesting

Author Response

I am very grateful to the reviewer for working with my article. I have substantially corrected the style and English grammar of the article.

I tried to make the order of presentation of the material more clear and supplemented the final part of the article.

Reviewer 3 Report

Review: Hidden person makes dialogue present. The place of IT in the system of dialogue according to Cohen, Buber, and Rosenzweig.

The essay begins with Rosenzweig’s famous critique of Buber’s supposedly impoverished sense of I-It in his dialogical system to develop, via Sigmund Freud and the rarely studied mystical tradition of Zadok Ha-Cohen (whose work predated Buber’s, but is highly relevant), a robust philosophy of dialogue that includes the third person as co-addressed by the second and first. As a philosophical project, the essay has much merit and reveals an excellent command of the works of Cohen, Buber, and Rosenzweig. Especially Cohen looms large, and the authors give a very helpful account of the dialogical elements in Cohen’s Ethik des reinen Willens and Religion der Vernunft. That alone gives the essay a valuable place in scholarship. The earlier works of Steven Schwarzschild and Trude Weiss Rosmarin (who once argued that Buber borrowed pretty much his entire Dialogphilosophie from Cohen) might be helpful addition here, as would be later authors, such as Dana Hollander.

Overall, the essay strikes me a strong piece with a real philosophical intervention into Buber’s I-you/I-it dichotomy, which also shines light onto Rosenzweig’s description of an it-like plastic cosmos. The richness of connections—including the phenomenology of the created thing with respect to Heidegger’s Da-sein—is impressive and exciting.

The immediate question that arises, however, is the place of Levinas in this conversation, especially as the authors seem to develop an inherent critique of Buber’s exclusivity of the I-You that resembles Levinas’s. Here, a brief excursion on Levinas on Buber would be helpful to better situate the author’s own contribution to the third-person presence in dialogue. Also Rosenstock-Huessy (with whom the authors engage in their essay) could deserve a more prominent place. Rosenstock’s critique of Buber dialogical temporality seems to me relevant here, as is the question of time altogether. Rosenstock’s pluri-aged self complicates the seemingly timeless, pure Gegenwart nature of the I-You/Thou in a way that mirrors the pluri-personal direction developed by the authors. I would encourage the authors to reach into this potential (i.e. the discussion of time), which actually nicely complements the suggestive remarks at the essay’s conclusion that the grammatical copula “is” indicates mere existence without the possibility of being addressed. There is a long tradition in modern Jewish thought, based on Hebrew linguistics, to question the indicative “is” and introduce elements of futurity that suggest both process and ethical meaning. (an early example would be S. H. Bergmann’s famous essay Kiddush Ha-shem of 1914 [?])

Author Response

I am very grateful to the reviewer for working with my article. I have substantially corrected the style and English grammar of the article.

I tried to make the order of presentation of the material more clear and supplemented the final part of the article.

I am very grateful to the reviewer for drawing my attention to the work of Steven Schwarzschild, Trude Weiss Rosmarin and Dana Hollander. Two monographs of Dana Hollander turned out to be especially important for me, and the I made reference to the first one in the article.

I am also very grateful to the reviewer for bringing to my attention the importance of the 3rd person concept and the asymmetry of interpersonal relationships in works of Levinas. Therefore, I have made additions to the final section of the article. However, the discussion about the 3rd person in Levinas’s works requires a separate paper, which can be further development of this article. This also explains the brevity of my analysis of the Rosenstock-Huessy idea. A detailed discussion of this important topic would require a substantial expansion of the article.

Once again, I am very grateful to the referee for the fruitful inspiring comments, which allowed not only to improve this article, but also to start working on subsequent ones.